

# Deep context-attentive transformer transfer learning for financial forecasting

Ling Feng and Ananta Sinchai

School of Integrated Innovative Technology, King Mongkut's Institute of Technology Ladkrabang, Bangkok, Thailand

## ABSTRACT

This study presents 2CAT (CNN-Correlation-based Attention Transformer), a deep learning model for financial time-series forecasting. The model integrates signal decomposition, convolutional layers, and correlation-based attention mechanisms to capture temporal patterns. A transfer learning framework is incorporated to enhance generalization across markets through pretraining, encoder freezing, and fine-tuning. Evaluation on six stock indices—Dow Jones Industrial Average (DJIA), Nikkei 225 (N225), Hang Seng Index (HSI), Shanghai Stock Exchange (SSE), Bombay Stock Exchange (BSE), and the Stock Exchange of Thailand (SET)—demonstrates strong predictive accuracy. On DJIA, 2CAT records an MSE of 0.0655, MAE of 0.2023, and $R^2$ of 0.9169, outperforming Deep-Transformer, which yields an MSE of 0.1360 and $R^2$ of 0.8274. The SET index, which posed challenges for previous models, demonstrates notable improvement with 2CAT, achieving an $R^2$ of 0.9094. Wilcoxon signed-rank test confirms statistically significant gains in non-transfer learning scenarios at the 0.05 level. Transfer learning experiments reveal statistically significant improvements, reinforcing the feasibility of cross-market knowledge transfer. An ablation study highlights the impact of architectural refinements and rotary positional encoding, while prediction horizon analysis confirms stable forecasting performance. These results establish 2CAT as a robust financial forecasting framework adaptable to diverse market conditions.

## INTRODUCTION

Stock indices serve as vital financial benchmarks, influencing derivatives such as options exchange traded funds (ETFs) with global trading volume surpassing USD 40 trillion by 2023 (*Zhu, 2023*). Accurate forecasting enables investors to mitigate risks and optimize strategies, while governments play a crucial role in stabilizing markets (*Jiang et al., 2020*; *Chen et al., 2022*). Predictions rely on historical data, market trends, and external factors to navigate financial volatility (*Xie, Rajan & Chai, 2021*). Traditional models like autoregressive integrated moving average (ARIMA) are effective for linear trends but struggle with nonlinear stock index behaviors (*Khan & Alghulaiakh, 2020*). Machine learning (ML) techniques, including support vector machines (SVM) and random forest (RF), offer improved predictive capabilities, outperforming conventional approaches (*Henrique, Sobreiro & Kimura, 2018*; *Zhang & Bai, 2019*; *Gunarto, Sa'adah & Utama, 2023*).

Corresponding author
Ananta Sinchai,
ananta.sin@kmitl.ac.th

Despite advancements, ML models depend on effective feature selection, which remains critical for enhancing forecasting accuracy across dynamic market conditions.

Modern deep learning techniques automate feature extraction, reducing dependence on manual engineering. While recurrent neural networks (RNNs) are effective for time-series forecasting, long short-term memory (LSTM) networks improve long-range dependency modeling (*Hochreiter & Schmidhuber, 1997*; *Gunarto, Sa'adah & Utama, 2023*). Hybrid models, such as LSTM-RNNs and convolutional RNNs (C-RNNs), further enhance predictive accuracy (*Ni et al., 2019*; *Li, Tian & Li, 2023*; *Smith et al., 2024*). Transformers, introduced by Google in 2017, outperform RNNs by leveraging self-attention (*Vaswani et al., 2017*). Transformer-based attention networks have been successfully applied to financial forecasting (*Zhang et al., 2022*). Building on these advancements, 2CAT is proposed to integrate machine learning to improve predictive performance in dynamic market environments.

Transfer learning (TL) reduces computational costs by enabling knowledge reuse across different tasks and datasets, even when they are not directly related (*Iman, Arabnia & Rasheed, 2023*). In stock price forecasting, TL has demonstrated that insights from one stock can enhance predictions for others (*He, Pang & Si, 2019*). However, industry-specific datasets yield better results than unrelated sectors, highlighting both the potential and challenges of TL in financial applications.

## Problem statements and literature review

Machine learning and deep learning have made significant strides in stock index price prediction, yet several challenges remain. This research addresses the following key issues:

(1) Stock market price movements are inherently nonlinear and nonstationary, meaning their statistical properties, such as mean and variance, fluctuate over time. Effective forecasting models must balance short-term dynamics with a broader understanding of long-term trends. In addition, stock prices exhibit cyclical and seasonal patterns, adding complexity to predictions. This study focuses on financial time-series data to explore forecasting methods that address these temporal challenges.

(2) Global financial markets are increasingly interconnected due to advancements in information dissemination, corporate crossholdings, and international market influences. While these factors shape broader trends, this study focuses on the individual behavior of stock index prices, setting aside complex interdependencies between markets.

Conventional stock price forecasting relies on mathematical models such as ARIMA and generalized autoregressive conditional heteroskedasticity (GARCH), which support volatility analysis and trend identification (*Dadhich et al., 2021*). GARCH with mixed data sampling has been applied to enhance volatility forecasting in the Chinese stock market (*Yu & Huang, 2021*). While effective in structured environments, linear models struggle with nonlinearity and non-stationarity, limiting adaptability in dynamic markets. Machine and transfer learning address these limitations by capturing short-term fluctuations and long-term trends. Recent studies highlight nonlinear forecasting methods, with SVM and

RF improving prediction accuracy (*Henrique, Sobreiro & Kimura, 2018*). Least squares SVM (LSSVM) demonstrates strong forecasting performance but has high computational costs (*Gong et al., 2019*). RF models have been used for predicting clean energy ETFs, outperforming logistic regression methods (*Sadorsky, 2021*). To enhance financial forecasting, the proposed 2CAT model leverages deep learning architectures, adapting to complex market behaviors.

Deep learning (DL) enhances stock market forecasting by modeling complex nonlinear relationships through hierarchical structures (*Gupta & Katarya, 2021*). RNNs are effective for time-series prediction but suffer from vanishing or exploding gradients. LSTM networks address this issue using memory cells and gating mechanisms, improving retention of relevant patterns (*Mehtab, Sen & Dutta, 2021*). Gated recurrent units (GRUs) further enhance training efficiency and enable feature extraction from industry-specific datasets (*Chung et al., 2014*; *Chen, Xue & Xing, 2023*). However, both LSTMs and GRUs face memory constraints when handling long historical sequences.

Convolutional neural networks (CNNs) provide strong feature extraction capabilities and improve stock trend prediction. Hybrid CNN-LSTM and CNN-BiLSTM-Attention models refine forecasting accuracy by integrating spatial and sequential patterns (*Lu et al., 2020*; *Zhang, Ye & Lai, 2023*). CNNs have also been used with attention mechanisms and GRUs in smart grid applications (*Li, 2023*). Despite their benefits, hybrid models increase computational complexity and pose risks of overfitting.

The transformer model captures long-range dependencies and enables parallel processing, making it effective for financial time-series forecasting (*Vaswani et al., 2017*). Originally developed for natural language processing, it has been adapted for stock prediction by integrating social media data with market trends (*Zhang et al., 2022*). Studies show transformers outperform traditional methods for major indices like CSI 300 and S&P 500, though they require large datasets and may overfit smaller financial data (*Wang et al., 2022*).

TL reduces training time and enhances financial forecasting, particularly in data-scarce environments such as emerging markets or unstable economic conditions. By leveraging insights from mature markets, TL improves predictive accuracy forecasting models (*He, Pang & Si, 2019*). Deep learning applications have integrated TL with industrial chain data using multilayer perceptron (*Wu, Wang & Wu, 2022*). Selective TL with adversarial training has demonstrated efficiency across datasets (*Li, Dai & Zheng, 2022*), while TL-based regression techniques refine accuracy and optimize transaction frequency (*Merello et al., 2019*).

Previous studies have successfully applied time-series forecasting models to financial datasets, demonstrating their ability to identify predictive patterns in stock trends. While research has explored short-term dynamics, long-term trends, and cyclical patterns, their simultaneous integration remains underexamined. Existing approaches often focus on specific markets or industries, overlooking broader market interactions that could enhance predictions. These limitations present an opportunity to develop a novel solution that aligns with the research objectives and contributions of this study, which will be outlined in the following section.

## Research objectives and contributions

This section presents the research objectives and contributions to clarify the focus of this study. To address the problem statements outlined in the previous section, the research objectives are as follows:

- Enhance the ability to capture long-term, short-term, and cyclical patterns in time-series data.
- Utilize interconnections across domains to improve prediction accuracy through TL.
- Develop a TL model that ensures robust generalization for unseen datasets.

To achieve these objectives, the key contributions of this study include:

1. Proposing a TL model incorporating 2CAT for multiday prediction.
2. Designing a CNN-correlation-based attention mechanism to extract short-term patterns, long-term dependencies, and cyclical similarities in time-series data.
3. Applying TL within the 2CAT framework to improve prediction accuracy for unobserved datasets.
4. Validating the effectiveness of 2CAT through experiments on six public stock indices across different timeframes.

## METHODS

This section presents the methodological approach for TL using the proposed 2CAT model, an enhanced transformer framework. Based on the conventional structure introduced by *Vaswani et al. (2017)*, 2CAT integrates CNN-based correlation mechanisms to improve time-series feature extraction, as illustrated in Fig. 1. The encoder and decoder modules are exhibited in Figs. 1A and 1B, respectively, incorporate key architectural modifications tailored for TL. Implementation begins with integrating signal processing techniques into the original transformer design. Following this, the procedural steps within the encoder and decoder modules are detailed, highlighting their distinct roles in pattern recognition. Finally, a training strategy for optimizing 2CAT in TL applications is introduced.

### Conceptual overview of the proposed 2CAT technique

Before delving into the mathematical details, it is important to establish a high-level understanding of our approach and its significance for time-series forecasting. This overview aims to make the technical content that follows more accessible to researchers across various disciplines. Time-series data presents unique challenges compared to other types of sequential data. While standard transformer models excel at capturing dependencies in sequential data like text, they were not specifically designed to handle time-series characteristics such as trends, seasonality, and cyclical patterns that occur at various time scales. Our approach transforms the standard transformer architecture by incorporating signal processing techniques familiar to researchers in time-series analysis.

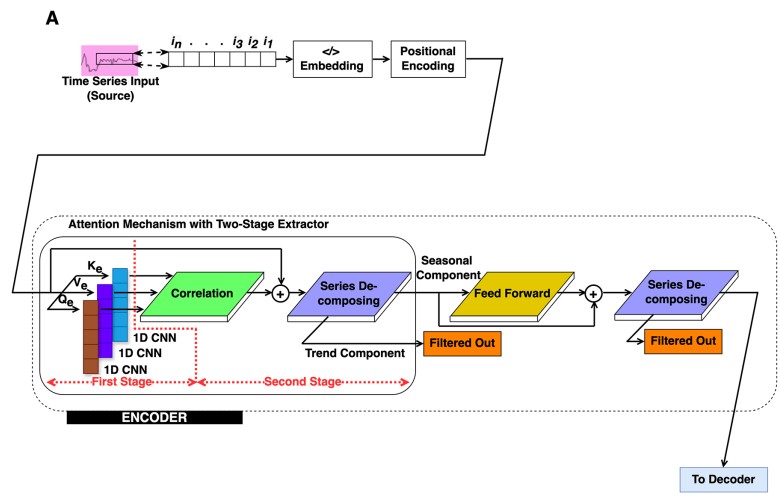

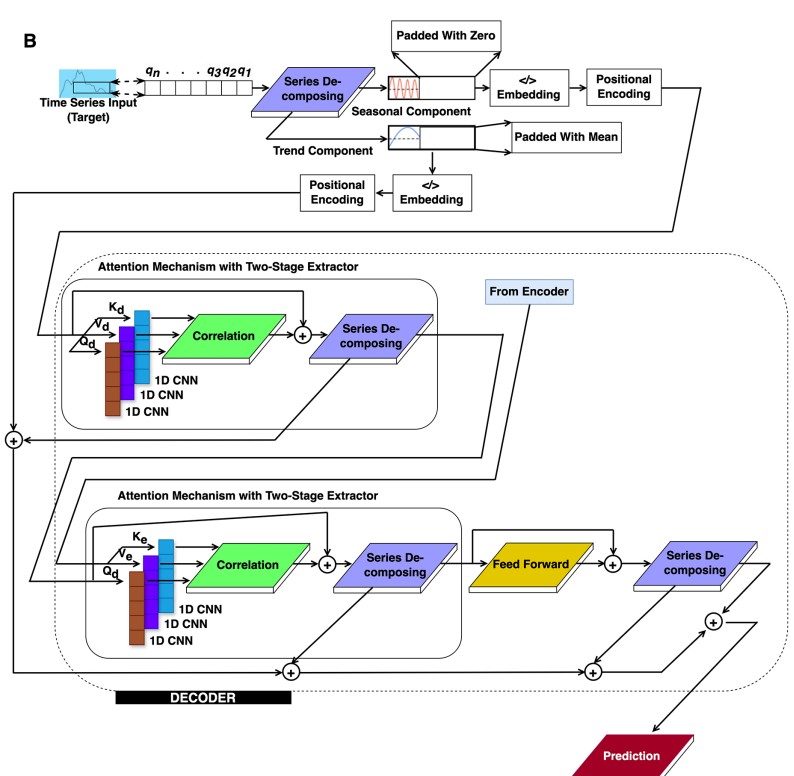

**Figure 1 The proposed structure.** (A) Encoder and (B) Decoder.

These modifications create a specialized model that better captures the temporal dynamics inherent in time-series data:

1. Breaking down the signal (series decomposing block): We decompose the input time-series into trend and seasonal components—a fundamental technique in

time-series analysis. This decomposition allows the model to process long-term trajectories and recurring patterns separately, similar to how analysts might manually separate these components when analyzing financial or climate data.

2. Local pattern recognition (1D CNN Block): We replace the standard linear projections with one-dimensional convolutional neural networks (1D CNNs). This modification helps identify localized patterns—such as sharp rises, falls, or fluctuations—that often contain critical information in time-series data like stock prices, sensor readings, or physiological signals.

3. Finding temporal relationships (correlation block): Instead of the standard attention mechanism, we implement correlation-based approaches that are particularly effective for time-series:

   ○ Autocorrelation: This detects how a time series relates to itself at different time lags, helping identify cyclical patterns such as daily, weekly, or seasonal effects.

   ○ Cross-correlation: This captures relationships between input and output sequences across various time shifts, essential for forecasting where current values may influence future outcomes with varying delays.

4. Optimal time lag detection (time-shift and concatenation): This mechanism identifies the most informative time delays for prediction—similar to how economists might determine that a leading indicator precedes economic changes by a specific number of months.

### Signal processing techniques modified in standard transformer model

To implement the conceptual approach described above, we modified the original transformer framework by introducing two key architectural enhancements within the proposed 2CAT model. First, we added a series decomposing block that separates time-series data into its fundamental components. Second, we replaced the traditional attention mechanism with a correlation block and a 1D CNN block, enabling a more effective capture of temporal relationships in time-series data. The correlation block serves different functions depending on its location: in the encoder, it performs autocorrelation to identify patterns within a single sequence; in the decoder, it performs cross-correlation to identify relationships between different sequences. Figure 1 illustrates this modified CNN-based correlation transformer architecture, with the encoder and decoder modules outlined by dashed lines.

### One-dimensional convolutional neural network

Traditional transformers use linear projections to create query, key, and value vectors. Instead, our model employs one-dimensional convolutional neural networks (1D CNNs) for this purpose. These 1D CNNs are particularly effective at capturing repeating short-term patterns in time series data.

After embedding and positional encoding, we process the input sequence through three parallel 1D CNNs with different parameter sets. These networks produce three distinct vector representations:

- $Q_\Delta$ (query vectors)
- $K_\Delta$ (key vectors)
- $V_\Delta$ (value vectors)

The $\Delta$-symbol indicates that these parameters belong to either the encoder ($\Delta = e$) or the decoder ($\Delta = d$) components. Each CNN applies standard convolutional operations to extract temporal features at different scales. These projections are then passed to the correlation block for further processing.

This 1D CNN approach replaces the standard linear projection layers found in the original transformer architecture, allowing our model to better identify and leverage temporal patterns in the data.

### Series decomposing

Time series data often contains multiple patterns operating at different scales. Series decomposition separates the input signal into two key components: seasonal patterns and underlying trends. This separation helps our model process each component more effectively. We use a simple moving average technique to extract the trend component, mathematically expressed Eq. (1).

$$x_{tc} = \frac{1}{N} \sum_{n=0}^{N-1} (x_n, k).$$  (1)

From Eq. (1), $x_{tc}$ represents the trend component calculated by averaging the input signal $x_n$ with stride $k$. We then isolate the seasonal component, $x_{sc}$, by subtracting the trend from the original input, $x$, by Eq. (2).

$$x_{sc} = x - x_{tc}.$$  (2)

This decomposition allows our model to process recurring patterns and long-term trends separately, improving prediction accuracy.

### Auto- and cross-correlation mechanisms in encoder and decoder

Our model improves upon the standard attention mechanism by incorporating signal processing techniques that capture temporal relationships more effectively. We implement these correlations in the frequency domain for computational efficiency.

To integrate autocorrelation and cross-correlation, the conventional attention mechanism was modified (Fig. 2). Both correlations are performed in the frequency domain to reduce computational complexity in handling $Q_\Delta$ and $K_\Delta$. The fast Fourier transform (FFT) enables this transformation.

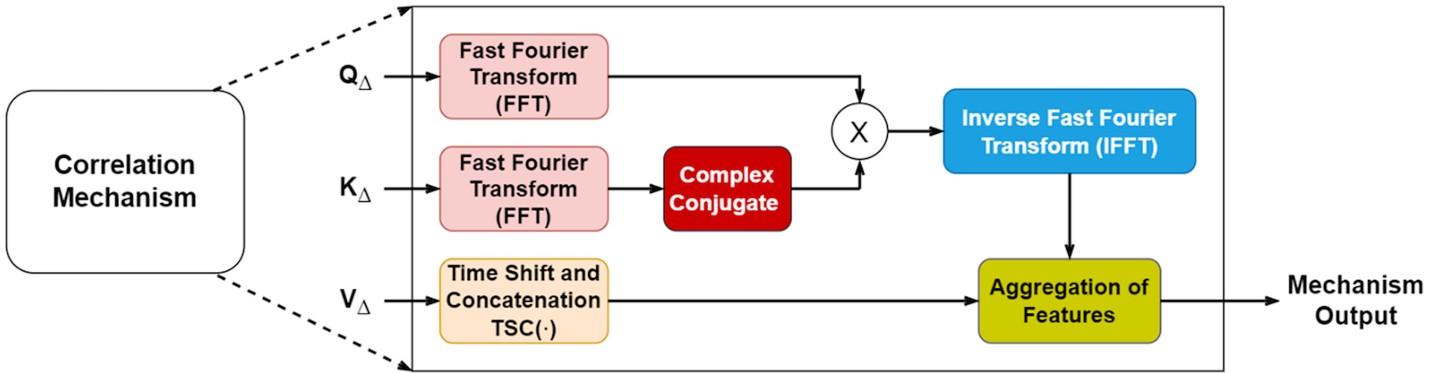

**Figure 2 Mechanisms of correlation.** Derived from *Feng & Sinchai (2025)*.

In the encoder, $Q_e$ undergoes FFT to produce $Q_e^F$. Its complex conjugate $Q_e^{*F}$ replaces $K_e$, and their multiplication results in autocorrelation, expressed in Eq. (3). This is then converted back to the time domain *via* inverse FFT, leading to Eq. (4), where $r_{Q_e Q_e}$ represents the autocorrelation.

$$R_{Q_e^F Q_e^{*F}} = Q_e^F Q_e^{*F} \tag{3}$$

$$r_{Q_e Q_e} = \text{IFFT}\left(R_{Q_e^F Q_e^{*F}}\right). \tag{4}$$

In the decoder, autocorrelation follows the process used in the encoder. Knowledge transfer occurs by passing $K_e$ and $V_e$ vectors. Cross-correlation is obtained by multiplying $Q_d$ and $K_e$ in the frequency domain, as shown in Eq. (5). Applying IFFT converts the cross-correlation back to the time domain, yielding Eq. (6), where $c_{Q_d K_e}$ represents the cross-correlation.

$$C_{Q_d^F K_e^{*F}} = Q_d^F K_e^{*F} \tag{5}$$

$$c_{Q_d K_e} = \text{IFFT}\left(C_{Q_d^F K_e^{*F}}\right). \tag{6}$$

The function $\text{TSC}_V(\cdot)$ manipulates $V_\Delta$ in both the encoder and decoder through time shift and concatenation (*Wu et al., 2021*). $V_\Delta$ shifts left by $m$, with the truncated section appended to the end, as shown in Fig. 3, where $N$ represents the data length. The green box marks the truncated part, which is reattached to the end, highlighted in magenta. Equation (7) defines the time length, linked to feature count, where $j$ is the integer result of $b$ times the logarithm of $N$, with $b$ as a hyperparameter.

$$j = \lfloor b \times \log N \rfloor. \tag{7}$$

Each shift is assigned a time length, denoted as $m_k$, where $k$ ranges from one to $j$. The accumulated length follows a consistent pattern: $m_2 = m_1 + m_1$, $m_3 = m_1 + m_2$, and the final length $m_k = m_1 + m_{k-1}$. This accumulation ensures $m_k$ remains less than $m_{k+1}$.

Autocorrelation and cross-correlation are computed using different values of $m_k$, where $k$ ranges from 1 to $j$, producing a series of correlation values. These values are sorted in descending order as $\tau_p$, with $\tau_1$ being the highest and $\tau_j$ the lowest. The sorting process is defined by the argTop($\cdot$) function in Eq. (8).

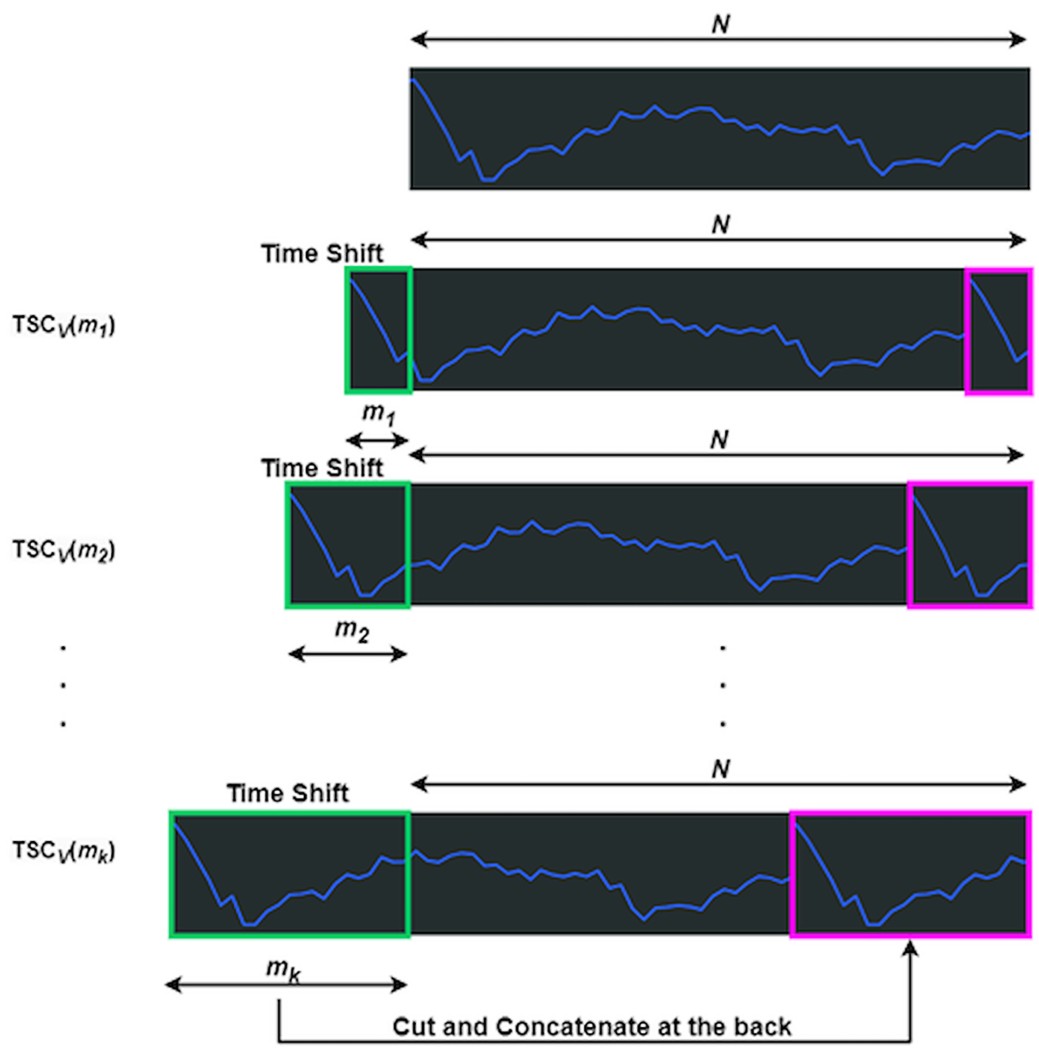

**Figure 3 Processes of a TSCᵥ function.** Derived from *Feng & Sinchai (2025)*

$$\tau_1, \tau_2, \ldots, \tau_j = \text{argTop}\left(r_{Q_\Delta Q_\Delta}(m_1), r_{Q_\Delta Q_\Delta}(m_2), \ldots, r_{Q_\Delta Q_\Delta}(m_j)\right). \tag{8}$$

To simplify the explanation, autocorrelation $r_{Q_\Delta Q_\Delta}$ is expressed, while cross-correlation replaces it with $c_{Q_\Delta K_\Delta}$. After deriving indices $\tau_1$ to $\tau_j$ *via* Eq. (8), $r_{Q_\Delta Q_\Delta}$ is approximated using a Softmax function, as shown in Eq. (9), where $\hat{r}_{Q_\Delta Q_\Delta}$ represents the approximation.

$$\hat{r}_{Q_\Delta Q_\Delta}(\tau_1), \hat{r}_{Q_\Delta Q_\Delta}(\tau_2), \ldots, \hat{r}_{Q_\Delta Q_\Delta}(\tau_j) = \text{Softmax}\left(r_{Q_\Delta Q_\Delta}(\tau_1), r_{Q_\Delta Q_\Delta}(\tau_2), \ldots, r_{Q_\Delta Q_\Delta}(\tau_j)\right) \tag{9}$$

After obtaining Eq. (9), feature aggregation is performed to derive autocorrelation and cross-correlation attention. Autocorrelation attention is calculated by summing the product of $\text{TSC}_V(\cdot)$ and $\hat{r}_{Q_\Delta Q_\Delta}(\cdot)$ in Eq. (10). Similarly, cross-correlation attention is the summation of $\text{TSC}_V(\cdot)$ and $\hat{c}_{Q_\Delta Q_\Delta}(\cdot)$, as shown in Eq. (11).

$$\text{Autocorrelation}-\text{Attention} = \sum_{k=1}^{j} \text{TSC}_V(\tau_k)\hat{r}_{Q_\Delta Q_\Delta}(\tau_k) \tag{10}$$

$$\text{Cross}-\text{correlation}-\text{Attention} = \sum_{k=1}^{j} \text{TSC}_V(\tau_k)\hat{c}_{Q_\Delta K_\Delta}(\tau_k). \tag{11}$$

## Work procedures of 2CAT

The 2CAT framework follows a structured sequence to process time-series data effectively. The procedure consists of the following key stages:

### Sequence decomposition

- Extracts trend and seasonal components from the input time series.
- These components are reserved for the decoder rather than directly used by the encoder.

### Embedding and positional encoding

- Converts raw time-series data into an embedded representation.
- Encodes positional information to retain temporal order.

### Encoder process

CNN-autocorrelation:

- After embedding and encoding, the data is passed to the encoder layers.
- The CNN-autocorrelation module captures long-range dependencies within the sequence.

  Sequence decomposing 1:

- Applies the first decomposition after a residual connection to refine feature extraction.

  Feedforward:

- Data is processed through a feedforward network, composed of convolutional layers, linear transformations, and GELU activation functions.

  Sequence decomposing 2:

- The output from the feedforward layer is combined with the first decomposition results.
- A second decomposition is performed to enhance time-series structure extraction.

### Decoder process

CNN-autocorrelation:

- Seasonal components enter the decoder, undergoing autocorrelation processing.

Sequence decomposing 1:

- Applies initial decomposition to refine seasonal features before further transformations.

CNN-cross-correlation:

- Computes cross-correlation between the decoder sequence and the encoder output.

Sequence decomposing 2:

- Conducts a second time-series decomposition, merging key components for improved representation.

Feedforward:

- Like in the encoder, data is processed through convolutional layers, linear transformations, and GELU activations.

Sequence decomposing 3:

- Combines feedforward outputs with the second decomposition results.
- Performs a third decomposition to further refine extracted patterns.

Aggregation:

- Trend components from multiple decomposition stages are aggregated.
- A final convolutional layer is applied for prediction, integrating all extracted temporal patterns.

## Training approach for transfer learning

TL enables us to leverage knowledge from one domain to enhance performance in another, particularly valuable when working with limited target data. In our research, we apply TL to reduce the computational resources required and accelerate training compared to starting from scratch. TL and fine-tuning are fundamental techniques within deep learning that facilitate the transfer of knowledge from one problem to another (*Zhuang et al., 2021*).

The architectural framework of our proposed approach is depicted in Fig. 4. We denote the source dataset as $D_s$ and the target dataset as $D_t$. Our methodology follows a structured three-stage process designed to effectively extract features from $D_s$ and apply them to $D_t$. Table 1 provides a summary of these three stages, outlining their purpose and impact on the model learning process.

### Pretraining stage

In this initial phase, we train our model using the source dataset ($D_s$). This critical foundation allows the model to learn broader patterns and representations that exist within the source domain. By capturing these general features first, we establish a knowledge base that can later be specialized for our target task.

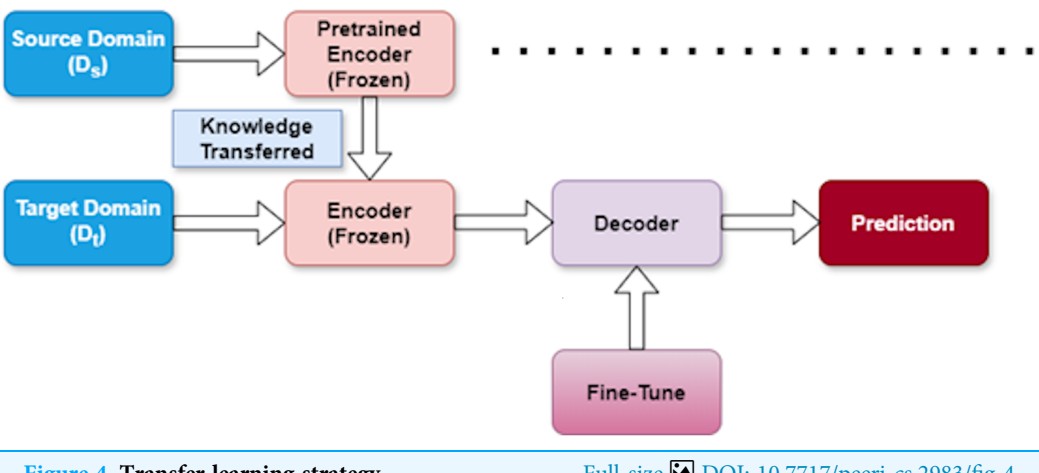

Figure 4 **Transfer learning strategy.**               

### Freezing stage

After the model has learned valuable representations from the source dataset, we enter the freezing stage. Here, we intentionally freeze the encoder layers within the model to preserve the knowledge acquired from $D_s$. This strategic freezing prevents these layers from being modified during subsequent training, ensuring that the valuable source domain knowledge remains intact. This preservation is essential for effective knowledge transfer between domains.

### Fine-tuning stage

In the final stage, we adapt the model to our specific target task using the target dataset ($D_t$). During fine-tuning, the model adjusts its parameters to align with the unique characteristics of the target domain. This calibration process improves the predictive capability of the model for the specific task while expanding on the knowledge transferred from the source domain.

This three-stage approach allows us to efficiently transfer knowledge between domains, reducing the need for extensive target domain data while maintaining strong performance. By systematically moving through pretraining, freezing, and fine-tuning, our model effectively bridges the gap between source and target domains.

## EXPERIMENT AND EVALUATION

### Dataset

This study was conducted on six stock index datasets: the US stock market and five other Asian stock markets. Daily trading information was retrieved from Yahoo Finance (https://finance.yahoo.com/). The data collection period spanned from February 1, 2021, to January 31, 2024. Each collected market dataset was stored in a 2D format, containing both text and numerical data, separated into five columns: trading date, opening price, highest price, lowest price, and closing price. Each price value belongs to the domain of real numbers ($\mathbb{R}$). Each collected 2D dataset is equivalent to a matrix of size $N \times 5$, where $N$ is the number of trading days in each market. Before processing, each market dataset was

**Table 1 The summary table showing stages, purposes, and effects.**

| Stage | Purpose | Effect |
|---|---|---|
| Pretraining | Learn general features from source dataset ($D_s$) | Establishes foundational knowledge from source domain |
| Freezing | Preserve learned representations | Prevents loss of valuable source knowledge during adaptation |
| Fine-tuning | Adapt to target dataset ($D_t$) | Optimizes model performance for the specific target task |

converted to a numeric dataset by eliminating the trading date column (text type). The resulting 2D numeric dataset was then transformed into a 1D dataset by serially concatenating the columns for opening price, highest price, lowest price, and closing price. The data are summarized in Table 2.

Following standard practice, all datasets were split into training, validation, and test sets in chronological sequence, adopting a 7:1:2 ratio. The computational experiments were conducted on a laptop equipped with a 1.70 GHz Intel Core i5-12500H CPU, an NVIDIA RTX 3050Ti graphics card, and 16 GB of RAM. The software frameworks included Keras 2.3.1 and PyTorch 1.13.1, running on a Python 3.9 interpreter. The proposed 2CAT model employs Transfer Learning (TL) to improve efficiency by leveraging pre-trained representations, reducing computational demands compared to full-scale training from scratch. To ensure scalability beyond consumer-grade hardware, the model was successfully deployed on Google Colab, executing without observed issues in a cloud environment. While this confirms functionality in cloud-based settings, further examination is necessary to assess its scalability in larger-scale financial applications, which may require enhanced computing resources, such as distributed systems or high-performance GPU clusters, to handle extensive datasets and real-time processing.

## State-of-the-art approaches in comparison

The TL model using the proposed 2CAT model was compared with three state-of-the-art (SOTA) approaches listed in Table 3. These SOTA models were chosen for their contemporary relevance and novelty. The study aimed to demonstrate that the TL model adopting the proposed 2CAT model is competitive with other SOTA techniques in the field of time-series forecasting. However, SOTA large language models (LLMs) were not included because they require extensive datasets. Testing them with only a few thousand data points would not fairly represent their capabilities and could impair their performance.

### CNN-BiLSTM-Attention

*Zhang, Ye & Lai (2023)* proposed a model based on CNN-BiLSTM-Attention to enhance the accuracy of predicting stock prices and indices. The baseline models in this study were based on the structure proposed by Zhang's architecture, but with different parameters. Initially, internal features were extracted using a CNN layer containing a 1D convolutional layer (instead of the 2D convolutional layer in the original model), a pooling layer, and a dropout layer. The BiLSTM layer processed these local features to identify patterns in dynamic internal changes. An attention mechanism was then employed to allocate varying

**Table 2 Datasets used in experiments.**

| Full name | Abbreviation | Sample size | Frequency |
| --- | --- | --- | --- |
| Dow Jones Industrial Average | DJIA | 755 × 4 = 3,020 | Daily |
| Nikkei 225 | N225 | 735 × 4 = 2,940 | Daily |
| Hang Seng Index | HSI | 738 × 4 = 2,952 | Daily |
| Shanghai Stock Exchange Composite Index | SSE | 729 × 4 = 2,916 | Daily |
| The S&P Bombay Stock Exchange Sensitive Index | BSE | 740 × 4 = 2,960 | Daily |
| Thailand SET Index | SET | 724 × 4 = 2,896 | Daily |

weights to the features detected by the BiLSTM, revealing deeper temporal relationships. The final output was processed using a dense layer.

### CNN-GRU-Attention

*Li (2023)* proposed a predictive model combining a CNN, attention mechanisms, and a GRU to optimize large-scale energy storage in smart grids. The baseline models in this study were based on *Li*'s *(2023)* structure but with different parameters. Initially, data were preprocessed and normalized to enhance the processing speed of CNN. The data were then fed into the CNN for feature extraction. As a 1D CNN was used, it was particularly suitable for time-series forecasting. After the flattening layer, the data were reduced dimensionally before passing through the GRU layer for further feature extraction, followed by an attention layer. The CNN-GRU structure was refined through autonomous weight distribution learning, ultimately leading to the predicted results.

### Deep-Transformer

*Wang et al. (2022)* validated the ability of the transformer to predict major global stock indices such as the CSI 300 and S&P 500. The baseline models in this study were based on Wang's structure, but with different parameters. The transformer architecture consisted of two encoders and one decoder, each featuring stacked layers of multi-heads self-attention and feed-forward layers.

### Informer

*Zhou et al. (2020)* proposed Informer, a Transformer-based model for long-sequence time-series forecasting. It enhances efficiency with ProbSparse self-attention and self-attention distilling, reducing computational complexity while preserving key dependencies. The generative-style decoder enables direct multi-step forecasting, making Informer well-suited for large-scale time-series applications.

### Experimental setup

All models were tested on the same six datasets containing daily historical trading data. The SOTA and proposed models utilized 10 d of historical data to forecast market prices for the subsequent 5 d. The predictive efficacy of the models was assessed using MSE, MAE, and $R^2$ metrics. The parameters of the proposed model are detailed in Table 3.

For the training parameters of the proposed model, the lowest, highest, open, and close prices were utilized as input data. The training function of the model was initiated by

**Table 3 Parameter setting for the proposed model and SOTA approaches.**

| Proposed model | Deep-Transformer | Informer | CNN-GRU-Attention | CNN-BiLSTM-Attention |
|---|---|---|---|---|
| Encoder layer = 2 | Encoder layer = 2 | Encoder layer = 2 | CNN layer = 2 | CNN layer = 1 |
| Decoder layer = 1 | Decoder layer = 1 | Decoder layer = 1 | GRU layer = 1 | BiLSTM layer = 1 |
| CNN layer = 1 | / | / | Attention layer = 1 | Attention layer = 1 |
| Number of heads = 4 | Number of heads = 4 | Number of heads = 4 | / | / |
| Seq-len = 10 | Seq-len = 10 | Seq-len = 10 | Seq-len = 10 | Seq-len = 10 |
| Pre-len = 5 | Pre-len = 5 | Pre-len = 5 | Pre-len = 5 | Pre-len = 5 |
| d_mode l= 512 | d_model = 512 | d_model = 512 | / | / |
| Loss function = MSE | Loss function = MSE | Loss function = MSE | Loss function = MSE | Loss function = MSE |
| Optimize = Adam | Optimize = Adam | Optimize = Adam | Optimize = Adam | Optimize = Adam |
| Batch size = 8 | Batch size = 8 | Batch size = 8 | Batch size = 128 | Batch size = 128 |
| Epoch = 30 | Epoch = 30 | Epoch = 30 | Epoch = 100 | Epoch = 100 |
| Learning rate = 0.0001 | Learning rate = 0.0001 | Learning rate = 0.0001 | Learning rate = 0.001 | Learning rate = 0.001 |

configuring these parameters. The dataset was divided into training, testing, and validation subsets with proportions of 70%, 20%, and 10%, respectively. Normalization was performed using a standard scalar, ensuring that each feature in the input sequence maintained a mean of zero and a standard deviation of one. The training phase was executed employing the Adam optimizer and MSE loss over 30 epochs. Strategies such as early stopping and learning rate adjustments were implemented to prevent overfitting and enhance model convergence.

Macroeconomic indicators, external events, and financial news are widely recognized as influential factors in financial forecasting. However, their direct incorporation into structured numerical models presents methodological challenges due to inconsistencies in data formats, reporting frequencies, and temporal resolutions. Sentiment analysis was considered as a potential enhancement, but the inherent misalignment between unstructured textual data and the structured nature of the current model rendered direct integration impractical.

While macroeconomic indicators such as GDP and inflation provide insights into broader economic conditions, financial time-series data primarily capture market sentiment and investor behavior, making them more responsive to short-term fluctuations. Given these methodological constraints, this study focuses exclusively on price-based financial modeling, utilizing structured historical trading data as its primary input rather than attempting broader economic forecasting. The exclusion of macroeconomic indicators stems from data compatibility challenges rather than a dismissal of their relevance in financial analysis.

To evaluate the generalization capabilities of the proposed model, TL was applied using the Dow Jones Industrial Average (DJIA) as the source dataset for pretraining the CNN-BiLSTM-Attention, CNN-GRU-Attention, deep-transformer, informer, and proposed 2CAT models. The five remaining Asian datasets served as target datasets, allowing assessment of model adaptability across different financial environments. During TL, the

encoder of the proposed 2CAT model was frozen, and only the decoder was fine-tuned to accommodate domain-specific features from the target datasets.

## RESULTS

### Attention mechanism analysis

Attention weight visualizations from five forecasting models—CNN-BiLSTM-Attention, CNN-GRU-Attention, Deep-Transformer, Informer, and the proposed 2CAT—are presented in Fig. 5. These visualizations provide insight into how each model interprets temporal dependencies in financial index data. CNN-BiLSTM-Attention (Fig. 5A) exhibits vertically stratified attention with emphasis on recent time steps, reflecting a strong temporal hierarchy where certain intervals consistently receive higher weights. In contrast, CNN-GRU-Attention (Fig. 5B) displays horizontal banding, suggesting uniform weighting across entire time steps, potentially indicating a sequential processing approach with limited point-to-point specificity. Deep-Transformer (Fig. 5C) introduces a semi-uniform distribution, predominantly weighted toward recent inputs, which, while less selective, allows for moderate differentiation across time points. Informer (Fig. 5D) further emphasizes temporal relationships through a gradient pattern, concentrating attention on the most recent data while progressively diminishing focus on earlier values, aligning with its design for sparse, long-sequence processing. Distinct from the others, 2CAT (Fig. 5E) reveals sparse and discontinuous attention patterns, marked by specific focal points rather than continuous regions, suggesting a prioritization of selective temporal interactions to capture precise dependencies rather than generalized trends. Taken together, these visualizations highlight structural differences in how each model allocates attention, shaping their ability to model financial time-series relationships with varying degrees of specificity and emphasis.

### Performance analysis of forecasting methods

The experimental results for forecasting methods applied to six stock market indices (Dow Jones Industrial Average (DJIA), Nikkei 225 (N225), Hang Seng Index (HSI), Shanghai Stock Exchange (SSE), Bombay Stock Exchange (BSE), and the Stock Exchange of Thailand (SET)) were evaluated using mean squared error (MSE), mean absolute error (MAE), and coefficient of determination ($R^2$). The analysis compared four established methods—CNN-BiLSTM-Attention, CNN-GRU-Attention, Deep-Transformer, and Informer—alongside the proposed 2CAT method, with transfer learning effects examined using DJIA as the source domain.

As shown in Table 4, the proposed 2CAT method consistently achieves lower error metrics across all indices. On DJIA, it records an MSE of 0.0655, MAE of 0.2023, and $R^2$ of 0.9169, outperforming Deep-Transformer (MSE: 0.1360, MAE: 0.2850, $R^2$: 0.8274). Similar advantages appear for HSI, where the proposed 2CAT method achieves MSE of 0.0146, MAE of 0.0945, and $R^2$ of 0.8212.

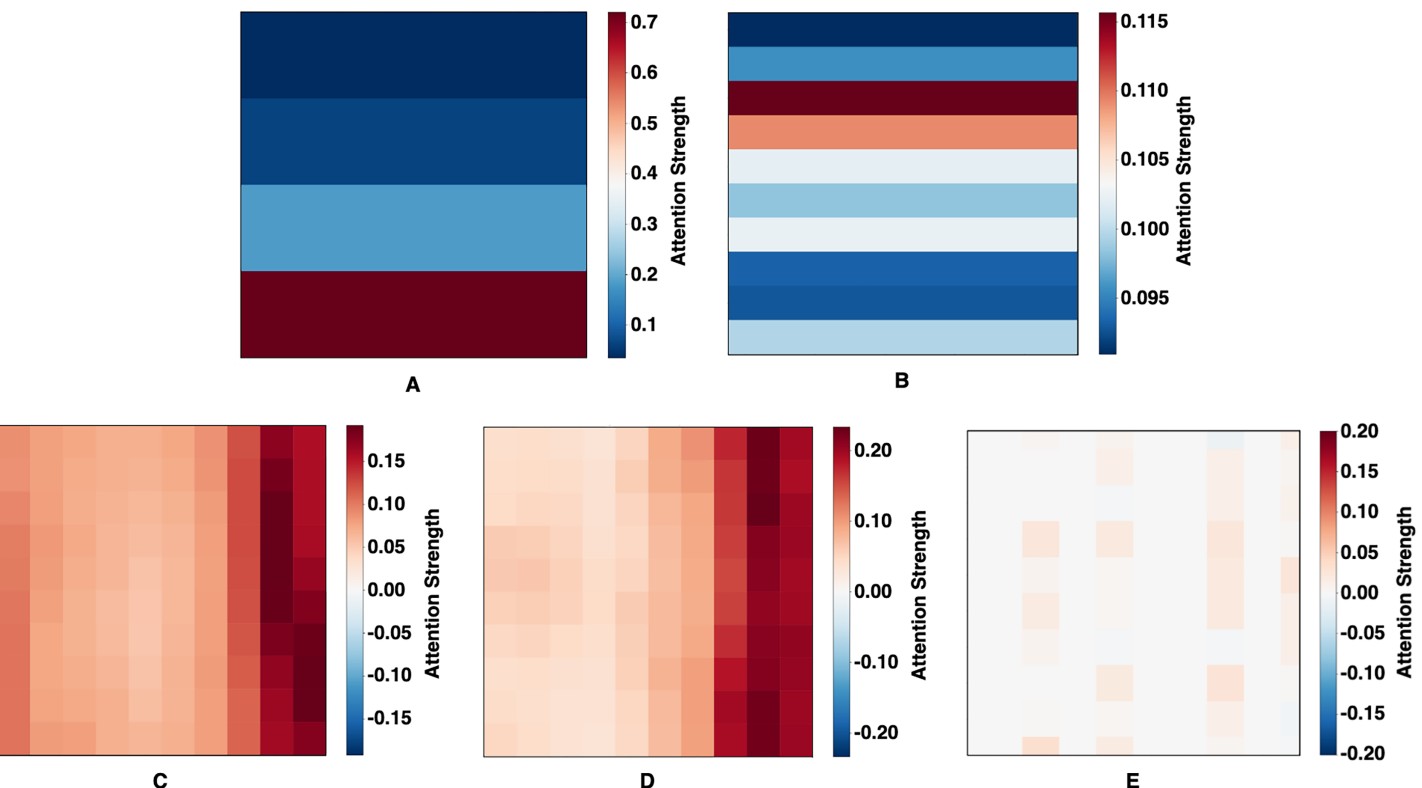

**Figure 5 Learned attention maps.** (A) CNN-BiLSTM-Attention, (B) CNN-GRU-Attention, (C) Deep-Transformer, (D) Informer, and (E) Proposed Technique (2CAT).

Performance gaps are most evident in SET, where the proposed 2CAT method (MSE: 0.1606, $R^2$: 0.9094) surpasses alternatives, which display negative $R^2$ values (−0.1656 to −1.5157). Deep-Transformer and Informer exhibit high volatility, particularly for N225 and BSE, indicating adaptation challenges across different market conditions.

## Impact of transfer learning

The results from transfer learning using DJIA as the source domain are presented in Table 5. The proposed 2CAT method maintains performance advantages, consistently outperforming benchmark models across target indices. On HSI, transfer learning improves MSE from 0.0146 to 0.0137, indicating effective knowledge transfer.

CNN-GRU-Attention remains stable, while CNN-BiLSTM-Attention exhibits variability, especially on SET, where $R^2$ declines from −1.5157 to −2.1231. Deep-Transformer and Informer continue to struggle with N225 and BSE indices under transfer learning conditions.

Overall, TL yields mixed results, offering slight improvements for some indices while causing marginal performance reductions for others. These findings suggest that while DJIA serves as a valuable source domain, target indices retain unique characteristics that require further model adaptation.

**Table 4 Experimental results for different forecasting methods.**

| | CNN-BiLSTM-Attention | | | CNN-GRU-Attention | | | Deep-Transformer | | | Informer | | |
|---|---|---|---|---|---|---|---|---|---|---|---|---|
| | MSE | MAE | $R^2$ | MSE | MAE | $R^2$ | MSE | MAE | $R^2$ | MSE | MAE | $R^2$ |
| DJIA | 0.2684 | 0.4314 | 0.6862 | 0.1958 | 0.3687 | 0.7711 | 0.1360 | 0.2850 | 0.8274 | 0.2170 | 0.3485 | 0.7244 |
| N225 | 0.5876 | 0.5936 | 0.5157 | 0.5447 | 0.5914 | 0.5511 | 4.2208 | 1.8077 | −2.9330 | 7.5803 | 2.5652 | −6.0677 |
| HSI | 0.0417 | 0.1657 | 0.5559 | 0.0200 | 0.1163 | 0.7871 | 0.0204 | 0.1075 | 0.7488 | 0.0193 | 0.1108 | 0.7625 |
| SSE | 0.1901 | 0.3425 | 0.6205 | 0.1478 | 0.3001 | 0.7050 | 0.1016 | 0.2223 | 0.7291 | 0.1324 | 0.2572 | 0.6467 |
| BSE | 0.2556 | 0.4039 | 0.3418 | 0.1443 | 0.2983 | 0.6283 | 0.3780 | 0.4685 | 0.0949 | 1.1570 | 0.9050 | −1.7733 |
| SET | 5.4917 | 1.8806 | −1.5157 | 1.0139 | 0.8353 | 0.5356 | 2.0672 | 1.1632 | −0.1656 | 3.0542 | 1.4268 | −0.7227 |

| | Proposed method | | |
|---|---|---|---|
| | MSE | MAE | $R^2$ |
| DJIA | 0.0655 | 0.2023 | 0.9169 |
| N225 | 0.3081 | 0.4242 | 0.7129 |
| HSI | 0.0146 | 0.0945 | 0.8212 |
| SSE | 0.0457 | 0.1626 | 0.8782 |
| BSE | 0.0467 | 0.1692 | 0.8881 |
| SET | 0.1606 | 0.3189 | 0.9094 |

## Statistical tests and confidence intervals

To assess the effectiveness of the proposed 2CAT method, statistical analysis is conducted to compare its performance against baseline methods across multiple datasets. This section outlines the research question and hypotheses that guide the evaluation, using summarized performance metrics presented in Tables 4 and 5. These consolidated results form the basis for testing whether the proposed 2CAT method demonstrates a statistically significant improvement over baseline approaches.

### Research question

This study examines whether the proposed 2CAT method demonstrates statistically significant improvements in performance compared to baseline methods across multiple datasets.

### Hypothesis testing

- Null hypothesis ($H_0$): No significant difference exists in performance, as measured by MSE, MAE, and $R^2$, between the proposed 2CAT method and the baseline SOTA methods.
- Alternative hypothesis ($H_1$): The proposed 2CAT method achieves significantly better performance, characterized by lower MSE and MAE, and higher $R^2$, compared to the baseline SOTA methods.

### Statistical significance analysis and interpretation

To assess the proposed 2CAT method, statistical analyses were performed using the Wilcoxon signed-rank test ($\alpha < 0.05$) with one-tailed testing across six market indices. This

**Table 5 Experimental results for transfer learning.**

| | Transfer learning CNN-BiLSTM-Attention | | | Transfer learning CNN-GRU-Attention | | | Transfer learning Deep-Transformer | | | Transfer learning Informer | | |
|------|--------|--------|---------|--------|--------|--------|--------|--------|---------|--------|--------|---------|
| | MSE | MAE | $R^2$ | MSE | MAE | $R^2$ | MSE | MAE | $R^2$ | MSE | MAE | $R^2$ |
| N225 | 0.6994 | 0.6298 | 0.4934 | 0.5433 | 0.5930 | 0.5523 | 2.7616 | 1.3262 | −1.5733 | 8.2405 | 2.6898 | −6.6832 |
| HSI | 0.0415 | 0.1687 | 0.5575 | 0.0220 | 0.1199 | 0.7650 | 0.0217 | 0.1102 | 0.7336 | 0.0195 | 0.1085 | 0.7599 |
| SSE | 0.1796 | 0.3363 | 0.6415 | 0.1287 | 0.2775 | 0.7431 | 0.1145 | 0.2516 | 0.6948 | 0.1512 | 0.2788 | 0.5967 |
| BSE | 0.3786 | 0.5099 | 0.0249 | 0.1514 | 0.3030 | 0.6101 | 0.4914 | 0.5304 | −0.1767 | 1.1243 | 0.8943 | −1.6948 |
| SET | 6.8176 | 2.0732 | −2.1231 | 0.9174 | 0.8111 | 0.5797 | 1.9966 | 1.1520 | −0.1258 | 3.1636 | 1.4263 | −0.7845 |

| | Transfer learning Proposed method | | |
|------|--------|--------|--------|
| | MSE | MAE | $R^2$ |
| N225 | 0.3035 | 0.4242 | 0.7172 |
| HSI | 0.0137 | 0.0933 | 0.8321 |
| SSE | 0.0452 | 0.1641 | 0.8796 |
| BSE | 0.0428 | 0.1604 | 0.8974 |
| SET | 0.1566 | 0.3134 | 0.9117 |

non-parametric test was selected for its robustness in comparing paired samples without assuming a normal distribution, making it suitable for financial time-series data, which often exhibits non-normal characteristics. Given the limited sample size ($n = 5$), parametric assumptions might be unreliable. The results of the proposed 2CAT method were analyzed against each baseline method separately for non-TL and TL contexts, with detailed statistical findings summarized in Table S1 in the supplementary information document.

To facilitate the analysis, Python was used with libraries NumPy and SciPy, ensuring efficient computation of significance values and confidence intervals. The computed differences between the models serve as the basis for statistical evaluation, allowing an assessment of whether the observed improvements are statistically significant.

- Non-TL performance

Statistically significant improvements across all evaluation metrics were confirmed in Table S1 in the supplementary information document. Analysis of MSE, MAE, and $R^2$ (Table 4) consistently demonstrated the superiority of the proposed 2CAT method. The null hypothesis ($H_0$) was rejected for all baseline comparisons, with $p$-values below 0.05. Statistical tests indicate that the proposed 2CAT method significantly outperformed baseline models across six market indices. The 95% confidence intervals for these differences excluded zero, reinforcing the robustness of the observed improvements.

- TL performance

The results for the transfer learning scenario (Table 5) reveal a more nuanced outcome. As indicated in Table S1 in the supplementary information document, the statistical

analysis rejects the null hypothesis ($H_0$), demonstrating a statistically significant improvement ($p < 0.05$) for the transfer learning implementation. While the transfer learning variant consistently outperforms baseline models across multiple indices, the degree of enhancement relative to the non-transfer learning implementation is statistically significant. This suggests that knowledge transfer across markets can enhance forecasting accuracy beyond market-specific training.

## Ablation study

This section presents an ablation study to evaluate the contribution of key components in the proposed 2CAT model for financial market forecasting. The experiments systematically assess the impact of architectural improvements over the base transformer, the effect of rotary positional encoding, and performance across different prediction horizons. Each component is individually examined to quantify its contribution to the overall model performance.

### Effect of architectural improvements over base transformer

Experimental results in Table 4 show that the proposed 2CAT model consistently outperforms Deep-Transformer across all market indices. On DJIA, 2CAT achieves an MSE of 0.0655, MAE of 0.2023, and $R^2$ of 0.9169, improving upon the performance of Deep-Transformer, which recorded an MSE of 0.1360 and $R^2$ of 0.8274. The N225 index exhibits the most significant gap, with 2CAT (MSE: 0.3081, $R^2$: 0.7129) surpassing Deep-Transformer (MSE: 4.2208, $R^2$: −2.9330). In transfer learning (Table 5), 2CAT maintains its advantage, while Deep-Transformer struggles to generalize, particularly on N225, BSE, and SET indices.

### Effect of rotary positional encoding

A comparison of the proposed 2CAT model with and without rotary positional encoding, presented in Table S2 in the supplementary information document, demonstrates consistent performance improvements across all market indices. On DJIA, rotary positional encoding reduces MSE from 0.0655 to 0.0598 (8.7% improvement), MAE from 0.2023 to 0.1904 (5.9% improvement), and increases $R^2$ from 0.9169 to 0.9241. Similar enhancements are observed in N225 and SET indices.

In Table S3 in the supplementary information document, the analysis extends to transfer learning, where rotary positional encoding continues to improve performance across target domains. On N225, MSE decreases from 0.3035 to 0.2889, with $R^2$ increasing from 0.7172 to 0.7308. HSI shows the most significant relative gain, with $R^2$ rising from 0.8321 to 0.8396. These results highlight the effectiveness of rotary positional encoding in enhancing forecasting accuracy.

### Prediction horizon analysis

The performance of the proposed model across different prediction horizons (1-day, 3-day, and 5-day) is analyzed in Table S4 in the supplementary information document. Prediction accuracy declines as the forecast period extends, though the model maintains acceptable performance at the 5-day horizon. On DJIA, MSE rises from 0.0188 (1-day) to

0.0655 (5-day), while $R^2$ declines from 0.9771 to 0.9169. The DJIA and HSI indices exhibit greater resilience, with 5-day $R^2$ values above 0.82, whereas the N225 index shows more pronounced degradation, with $R^2$ decreasing from 0.9126 to 0.7129, highlighting variations in market predictability and efficiency.

The ablation study results demonstrate the substantial improvements achieved through architectural enhancements over the base transformer model, with the proposed 2CAT architecture consistently outperforming Deep-Transformer across all market indices. Additionally, the effectiveness of rotary positional encoding is evident in enhancing model performance for both direct prediction and transfer learning scenarios. Furthermore, the analysis of prediction horizons provides valuable insights into the temporal limitations of the forecasting capabilities, confirming that the proposed model maintains robust performance within practical forecasting timeframes.

## DISCUSSION

The ability to develop reliable financial forecasting models hinges on balancing adaptability with stability across diverse market conditions. The proposed 2CAT model addresses this challenge by integrating context-attentive architectures that refine predictive accuracy while maintaining robustness. This section examines its performance across multiple stock indices, highlighting key findings related to cross-market generalizability, model sensitivity, and the impact of transfer learning. In addition, theoretical and practical implications, as well as limitations and future directions are also discussed.

### Cross-market generalizability

The numerical findings are visually supported by the bar charts in Fig. 6. The proposed 2CAT method, represented by dark and light blue bars, consistently achieves lower MSE and MAE values across all indices. The $R^2$ chart highlights consistently higher values, indicating better explanatory power for stock price movements. While established models perform well on some indices, they lack robustness across markets. The proposed 2CAT method maintains stable performance, enhancing generalizability for stock market forecasting applications.

### Model sensitivity

Performance metrics reveal varying model sensitivity to market characteristics. Deep-Transformer and Informer show significant degradation on certain indices, such as N225, indicating adaptation challenges. In contrast, the proposed 2CAT method demonstrates stability across diverse conditions, suggesting improved robustness. Modest gains in transfer learning highlight difficulties in knowledge transfer across markets with different financial dynamics, emphasizing the need for models that balance generalizability with market-specific adaptability.

### Key insight on transfer learning

The findings confirm that the proposed 2CAT method delivers statistically significant improvements over existing forecasting approaches. Furthermore, the transfer learning implementation provides measurable advantages over the non-transfer variant, reinforcing

**MSE**

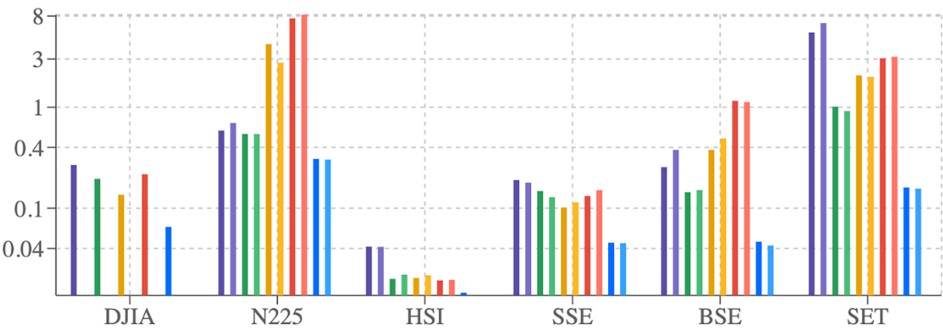

**MAE**

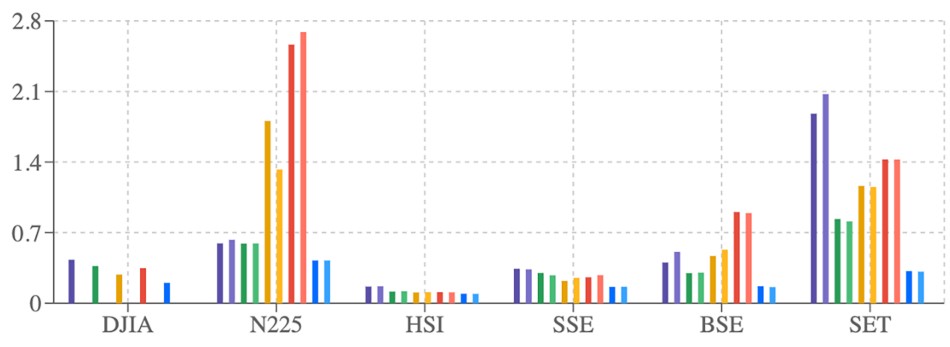

**R²**

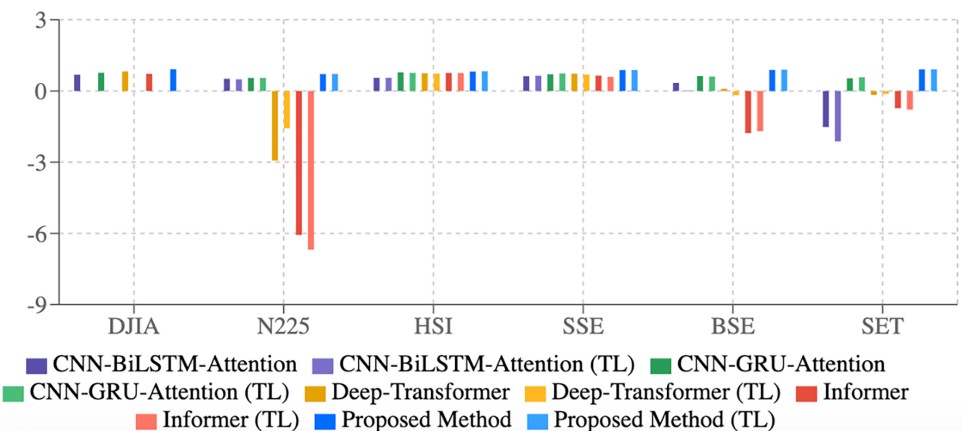

**Figure 6 Evaluation of model performance using multiple metrics.**

its potential for cross-market applications. These results, supported by statistical tests (Table S1), contribute to discussions on the efficacy of transfer learning in financial forecasting and highlight its relevance in multi-market modeling.

## Theoretical and practical implications

This study offers both theoretical and practical contributions to financial time-series forecasting. The proposed 2CAT model, built on a context-attentive transformer architecture, demonstrates that financial markets benefit from models capable of capturing complex temporal dependencies, including long-term trends, short-term fluctuations, and cyclical patterns. The integration of signal decomposition, sparse attention mechanisms, and convolutional layers enhances the ability of the model to reflect non-linear and non-stationary behaviors observed in real-world financial data. The analysis of attention weight distributions suggests that domain-specific attention patterns provide improved forecasting accuracy. This supports the argument that selective and context-aware attention structures are more suitable for financial applications than uniformly distributed attention approaches.

On the practical side, the experimental results show that 2CAT performs reliably across a range of international stock indices, including those with high volatility. Its strong performance on the SET index, where other models failed, highlights its adaptability in challenging market conditions. This robustness makes it suitable for risk management, investment planning, and strategy development tailored to specific market dynamics. Notably, transfer learning demonstrated statistically significant improvements in forecasting accuracy, reinforcing its potential as an effective approach for leveraging knowledge across structurally different markets. These results indicate that while careful adaptation remains necessary, transfer learning can enhance model efficiency and predictive performance in financial applications.

Given its consistent predictive performance across multiple forecast horizons, 2CAT is well-suited for implementation in financial decision support systems, especially in scenarios relying on structured historical price data.

## Limitations and future directions

The study lacks computational efficiency assessment and inference latency benchmarks across hardware configurations, critical for determining suitability in real-time financial applications. Guidelines for model retraining frequency in evolving markets are also absent. The architecture excludes sentiment analysis and external event integration, limiting predictive capacity to price-based signals. Preliminary experiments with macroeconomic indicators showed inconsistent correlations (Table S5), with temporal misalignment between economic and financial time-series data preventing their integration.

The South African market dataset was selected as an ideal case study for emerging market dynamics, avoiding high acquisition costs and licensing restrictions associated with more frequently studied markets. Its relative isolation from global financial centers allows examination of localized economic patterns with reduced external interference.

For comparative analysis, CNN-BiLSTM-Attention, CNN-GRU-Attention, Deep-Transformer, and Informer models were included. However, advanced architectures like

Time2Vec and Temporal Fusion Transformer were excluded due to implementation constraints, creating a gap in the comparative evaluation.

Future research will focus on computational efficiency through hardware benchmarking, developing adaptive retraining frameworks based on market volatility, and integrating structured financial data with sentiment analysis capabilities. The study will explore distributed computing implementations for real-time applications and model pruning techniques to optimize inference time. The implementation of excluded models will be considered once aligned financial and macroeconomic features are promptly implemented.

## CONCLUSIONS

This study presents 2CAT, a context-attentive transformer architecture developed for financial time-series forecasting. The model integrates series decomposition, convolutional layers, and correlation-based attention mechanisms to capture complex temporal patterns across multiple market indices.

Performance evaluation across six stock indices—DJIA, N225, HSI, SSE, BSE, and SET—demonstrates significant improvements over established models, with statistically validated enhancements in non-transfer learning settings. On the DJIA dataset, 2CAT achieved an MSE of 0.0655, MAE of 0.2023, and $R^2$ of 0.9169, outperforming Deep-Transformer, which recorded an MSE of 0.1360 and $R^2$ of 0.8274. The challenging SET index, where baseline models struggled to identify underlying patterns and yielded $R^2$ values as low as $-1.5157$, saw substantial improvement with 2CAT, which achieved an $R^2$ of 0.9094. Statistical validation *via* the Wilcoxon signed-rank test confirmed that enhancements in non-transfer learning scenarios were significant at the 0.05 level.

An ablation study examines key architectural components, including rotary positional encoding, which consistently contributes to forecasting accuracy. Analysis of different prediction horizons highlights the model's ability to maintain reliable performance across various forecast durations.

Transfer learning experiments reveal statistically significant improvements, reinforcing the feasibility of cross-market knowledge transfer. Overall, 2CAT provides a structured, context-aware framework for financial forecasting, demonstrating adaptability across diverse market conditions.

### Funding

This work was supported by King Mongkut's Institute of Technology Ladkrabang (No. 2567-02-10-002). The funders had no role in study design, data collection and analysis, decision to publish, or preparation of the manuscript.

## Grant Disclosures
The following grant information was disclosed by the authors:
King Mongkut's Institute of Technology Ladkrabang: 2567-02-10-002.

## Competing Interests
The authors declare that they have no competing interests.

## Author Contributions
- Ling Feng conceived and designed the experiments, performed the experiments, performed the computation work, prepared figures and/or tables, authored or reviewed drafts of the article, and approved the final draft.
- Ananta Sinchai conceived and designed the experiments, performed the experiments, analyzed the data, authored or reviewed drafts of the article, investigation, validation, supervision, and approved the final draft.

## Data Availability
The code and data are available at Zenodo: Feng, L., & Sinchai, A. (2025). Source code for Deep Context-Attentive Transformer Transfer Learning for Financial Forecasting. Zenodo. https://doi.org/10.5281/zenodo.15400641.

## Supplemental Information
Supplemental information for this article can be found online at http://dx.doi.org/10.7717/peerj-cs.2983#supplemental-information.

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
