# Peer review of "Deep context-attentive transformer transfer learning for financial forecasting"

_PeerJ Computer Science, doi:10.7717/peerj-cs.2983_

## Round 0.1 · original submission · Minor Revisions

Thank you for submitting your manuscript. In light of the reviewers’ comments, I invite a minor revision.

Please break up long sentences, especially in the Introduction and Methods, for greater clarity, add significance tests or confidence intervals for the reported gains, clarify how the CNN-correlation block plugs into the Transformer and transfer-learning stages, and either moderate or better substantiate the “economic forecasting” claim. A brief interpretability example and a code/data link would further enhance transparency.

Kindly resubmit with a short point-by-point response. I look forward to your revised manuscript.

Reviewer 1 ·

Basic reporting

According to the manuscript, we could observe that the main question addressed here is: How can financial time-series forecasting be improved to better capture nonlinear, nonstationary patterns and support effective transfer learning across different stock markets? Furthermore, the author investigates whether introducing a novel CNN-based correlation attention mechanism into a Transformer model, combined with transfer learning, can significantly enhance forecasting accuracy in financial-economic data compared to current state-of-the-art models.

Overall, the manuscript is well-constructed and written in professional English. The authors also provided the raw data and the results that are aligned with the hypotheses. However, I would have a minor suggestion, which is to simplify some overly long and complex sentences, especially in the Introduction and Methods sections. That would enhance the readability for the readers.

Experimental design

The paper originally falls well within the Aims and Scope of the journal. And the questions of the research are clearly defined, highly relevant, and address an important gap in financial-economic forecasting. The experimental design is rigorous, with detailed methodology. There is a potential improvement that could be to add more interpretability analysis. For example, the author could discuss the visualization of learned attention maps, which is to enhance the experimental transparency.

Validity of the findings

The findings are robust, solid, and well-supported by the data presented. The experimental comparisons with other SOTA models are meaningful and carefully discussed. However, it would be better if the author would limit their conclusions to what is supported by the results without overstating any other claims. Based on the information mentioned above, I would have a minor suggestion for the author, which is to consider conducting statistical significance tests between competing methods in future work to further reinforce claims of improvement.

Additional comments

The manuscript does make a valuable contribution by introducing a novel CNN-correlation attention mechanism within a Transformer framework for financial time-series forecasting, combined with a practical transfer learning approach.

I would give Minor revisions as my recommendation. Please review the suggestions mentioned above. That would enhance the overall quality of the paper.

Cite this review as

Reviewer 2 ·

Basic reporting

- The manuscript is clearly written and well-structured, adhering to the journal’s standards.
- Figures and tables are well-labeled, and raw data is appropriately shared.
- The literature is generally well referenced, although most of it focuses on technical modeling rather than broader financial-economic contexts.

Experimental design

The paper presents a novel hybrid architecture (2CAT) that combines CNN-based correlation mechanisms with a transformer backbone. The transfer learning (TL) framework is validated using six international stock market indices.

However, the dataset is composed solely of historical price data (open, high, low, close), which raises concerns about whether the study captures real-world external events, macroeconomic factors, or financial news — all of which are vital in financial-economic forecasting (Major concern). The stated objective to support “economic decision-making” is not well supported by the limited input features.

In addition, no SOTA attention-based transformer models like Informer, Time2Vec, or Temporal Fusion Transformer are included for comparison.

There is no ablation study to isolate the effect of the CNN-Based Correlation Layer vs. the Transformer alone.

Validity of the findings

The proposed model shows superior performance over selected baselines across several markets and metrics (MSE, MAE, R2).

However, again, the feature space is extremely narrow, consisting only of historical prices, which limits the generalizability and practical applicability of the results in realistic financial settings.

The authors claim applicability to broader economic indicators (e.g., GDP, inflation) without testing such data, which weakens the conclusions.

Additional comments

The paper does not discuss the computational efficiency or inference time of the proposed architecture, whether it is viable for real-time deployment, especially in low-latency financial environments, how often the model would be retrained or updated, which is important for time-evolving markets.

Cite this review as

Reviewer 3 ·

Basic reporting

The manuscript is well structured, professionally written, and generally adheres to the editorial standards of PeerJ Computer Science. The authors provide a detailed background review that situates their work within the context of existing research on stock index forecasting. The abstract effectively summarizes the objectives, methodology, and key findings.

The introduction presents a clear motivation for the study and a comprehensive literature review, citing both traditional statistical models and recent deep learning architectures. Figures and tables are clearly labeled and enhance the comprehension of results. The raw data is appropriately sourced from Yahoo Finance, but it would be beneficial if the authors provided a repository link to facilitate replication.

That said, several portions of the manuscript become overly technical without providing sufficient high-level framing, which may pose comprehension challenges to a broader scientific audience not deeply familiar with deep learning architectures. These include:

Section 3: Methods (Lines ~223–314): The explanation of the CNN-correlation block, auto/cross-correlation via FFT, and Equations (1)–(11) is heavily mathematical. While mathematically rigorous, these sections could benefit from a simplified conceptual summary before diving into technicalities—e.g., what each component is trying to achieve in the context of time-series forecasting.

Section 4.3: Training tactic for transfer learning (Lines ~349–380): The pretraining-freezing-finetuning sequence is well explained, but the reader would benefit from a high-level visualization or summary table highlighting the rationale and effect of each stage.

Experimental design

The authors introduce the 2CAT model, which combines CNN-based correlation attention mechanisms with a modified transformer architecture, and apply transfer learning (TL) to address the challenge of financial forecasting across multiple international markets. This dual innovation—enhancing attention mechanisms and leveraging TL across heterogeneous datasets—is timely and technically sound.

The experimental design is rigorous. The authors conduct multiple layers of evaluation, including comparisons against three strong baselines (CNN-BiLSTM-Attention, CNN-GRU-Attention, Deep-Transformer), an ablation test isolating CNN-correlation’s contribution, and forecasting across different temporal windows (1-, 3-, and 5-day horizons). These experiments are well thought out and convincingly demonstrate the strength of the proposed approach.

However, a few concerns remain:

The justification for using financial data as a proxy for economic forecasting is insufficient. The authors mention that “financial time-series data... serve as a suitable alternative” due to the unavailability of economic time-series data. While this may be practical, it is not conceptually justified. Financial time series (e.g., stock prices) reflect investor behavior and market sentiment and are often far more volatile and reactive than macroeconomic indicators like GDP or inflation. The authors should elaborate on the theoretical or empirical basis for this substitution—e.g., are there studies showing meaningful correlations between market indices and economic indicators over time? Without such justification, the extension of the model’s applicability to “economic forecasting” in the title and abstract may be overstated.

Validity of the findings

The findings are convincing and supported by comprehensive experiments. The 2CAT model consistently outperforms all benchmarks across six datasets, with notable improvements in MSE, MAE, and R². Transfer learning performance is particularly strong: the model pretrained on the DJIA demonstrates robust generalization to Asian markets.

Ablation tests validate the design choices, and time-scale analyses demonstrate temporal flexibility. However, the following improvements would strengthen the validity:

1) Statistical Significance: The authors report average metric improvements, but there is no indication of statistical testing (e.g., paired t-tests, confidence intervals). Given the relatively small number of target datasets, reporting whether improvements are statistically significant would add credibility.

2) Data Diversity and Model Generalization: While the use of six indices across different geographies is commendable, the model is still limited to stock data. Claims about generalizability to broader economic indicators are currently speculative and need empirical support.

Additional comments

Strengths:

1) Novel architecture combining CNN-correlation with a transformer and dual-stage decomposition.

2) Creative and practical use of transfer learning to extend forecasting capability to less data-rich markets.

3) Extensive experimental design including multiple datasets, baseline models, ablation study, and multi-horizon forecasting.

Areas for Improvement:

Clarify Financial vs. Economic Data Usage
The manuscript's title and abstract refer to “economic forecasting,” but the study uses only financial time series (stock indices). Although the authors state (Lines 91–95) that financial data are used due to the unavailability of economic data and reflect “similar temporal forecasting patterns,” this explanation is insufficient.

The authors should either:
a) Provide evidence or references justifying that stock indices are reliable proxies for broader economic indicators like GDP or inflation, or
b) Reframe the title and scope to focus strictly on “financial time-series forecasting.”

Clarify Complex Sections with More Intuition
The mathematical detail in the Methods section (especially Equations 1–11) is appreciated, but overly dense. Providing high-level intuition, diagrams, or pseudocode will make the work accessible to broader readers in financial computing and applied ML.

Supplement Statistical Evaluation
To reinforce result validity, please consider adding statistical significance tests (e.g., p-values, confidence intervals) comparing the proposed model against baselines, especially for transfer learning performance (n=5 target datasets). While limited in sample size, simple bootstrapping or Wilcoxon signed-rank tests would add credibility.

Improve Reproducibility and Data Transparency
The paper describes raw data sources well, but does not include code or a data repository. Providing code (e.g., GitHub) and preprocessed data would substantially increase transparency and replicability.

Clarify Transfer Learning Practicality
The TL procedure is clear, but training was performed on a consumer-grade laptop (Line 397). Please comment briefly on the scalability of the approach for larger-scale deployment in real-world financial services contexts.

Cite this review as

---

## Round 0.2 · accepted · Accept

Thank you for your thoughtful revisions and for addressing the reviewers' feedback in detail. I am pleased to inform you that your manuscript is now accepted for publication.

Reviewer 3 has recommended acceptance, highlighting the methodological rigor, clarity of exposition, and practical contributions of your 2CAT model. Reviewer 2 noted one remaining suggestion regarding the inclusion of traditional statistical models such as ARIMA and GARCH for comparison. While this would add further context, the strength of your empirical validation and the clarity of your scope make the current version suitable for publication.

Congratulations on your work, and we look forward to seeing it published.

Reviewer 2 ·

Basic reporting

Most of the concerns have been addressed, while I still have one comment for further improvement.

Experimental design

Since the authors only include OLHC prices as the input features (without indicators or news sentiment, etc.), the models then must be compared with traditional statistical/technical methods, e.g. ARIMA and GARCH, which have been mentioned in the literature review, but not included in the comparison.

Validity of the findings

No comment.

Cite this review as

Reviewer 3 ·

Basic reporting

Language & Clarity:
The manuscript is generally well-written in professional and clear English.

Background & Literature:
The introduction provides a comprehensive overview of relevant prior work and clearly articulates the knowledge gap the proposed model seeks to fill. References are relevant and up to date, including recent studies up to 2023.

Figures & Tables:
Figures are relevant, well-labeled, and support the claims made in the text. The attention visualization (Figure 5) is particularly helpful in contrasting model behavior.

Data Availability:
Raw data and code are appropriately shared via Zenodo, fulfilling PeerJ’s open data policy.

Experimental design

Originality & Relevance:
The 2CAT model integrates signal decomposition, CNNs, and correlation-based attention, which is novel and well-justified. The study is squarely within the journal’s scope.

Research Questions & Rationale:
The paper clearly identifies challenges in financial time-series forecasting and motivates the proposed 2CAT model as a solution. The inclusion of both traditional forecasting and transfer learning contexts adds robustness to the design.

Method Detail & Reproducibility:
The methods section is exceptionally detailed. All components of the architecture, including CNN layers, correlation mechanisms, and the TL strategy (pretraining, freezing, fine-tuning), are described with mathematical clarity. However, some algorithmic steps (e.g., hyperparameter settings) could be better summarized in a parameter table for quick reference.

Validity of the findings

Data Quality & Statistical Soundness:
Experiments span six diverse stock indices, improving external validity. The Wilcoxon signed-rank test is appropriately used to validate statistical significance. The use of confidence intervals and ablation studies strengthens the robustness of the conclusions.

Results Interpretation:
The results are interpreted accurately, and conclusions are tied to the research questions. The model’s improved performance, particularly on challenging datasets like SET, is well-supported.

Limitations:
The paper commendably discusses its limitations, especially regarding the absence of macroeconomic/sentiment data and computational efficiency considerations. This adds credibility to the findings.

Cite this review as